# Trade-Offs among Release Treatments in Jack Pine Plantations: Twenty-Five Year Responses

Holly D. Deighton [1,*], Frederick Wayne Bell [1], Nelson Thiffault [2], Eric B. Searle [1], Mathew Leitch [3], Mahadev Sharma [1] and Jennifer Dacosta [1]

[1] Ontario Forest Research Institute, Ontario Ministry of Natural Resources and Forestry, 1235 Queen St. East, Sault Ste. Marie, ON P6A 2E5, Canada; wayne.bell@ontario.ca (F.W.B.); eric.searle@ontario.ca (E.B.S.); mahadev.sharma@ontario.ca (M.S.); jennifer.dacosta@ontario.ca (J.D.)

[2] Canadian Wood Fibre Centre, Canadian Forest Service, Natural Resources Canada, 1055 du P.E.P.S., P.O. Box 10380, Sainte-Foy Stn., Québec City, QC G1V 4C7, Canada; nelson.thiffault@canada.ca

[3] Faculty of Natural Resources Management, Lakehead University, Wood Science Testing Laboratory, 955 Oliver Rd., Thunder Bay, ON P7B 5E1, Canada; mleitch@lakeheadu.ca

* Correspondence: holly.deighton@ontario.ca; Tel.: +1-519-274-3314

**Abstract:** We assessed 27 indicators of plant diversity, stand yield and individual crop tree responses 25 years post-treatment to determine long-term trade-offs among conifer release treatments in boreal and sub-boreal forests. This research addresses the lack of longer-term data needed by forest managers to implement more integrated vegetation management programs, supporting more informed decisions about release treatment choice. Four treatments (untreated control, motor-manual brushsaw, single aerial spray, and complete competition removal) were established at two jack pine (*Pinus banksiana* Lamb.) sites in Ontario, Canada. Our results suggest that plant diversity and productivity in boreal jack pine forests are significantly influenced by vegetation management treatments. Overall, release treatments did not cause a loss of diversity but benefitted stand-scale yield and individual crop tree growth, with maximum benefits occurring in more intensive release treatments. However, none of the treatments maximized all 27 indicators studied; thus, forest managers are faced with trade-offs when choosing treatments. Research on longer term effects, ideally through at least one rotation, is essential to fully understand outcomes of different vegetation management on forest diversity, stand yield, and individual crop tree responses.

**Keywords:** vegetation management; *Pinus banksiana*; plantations; herbicides; glyphosate; growth and yield; diversity

## 1. Introduction

International agreements such as the Paris Climate Agreement, the UN Sustainable Development Goals and the Montréal Process are providing opportunities and setting challenges for forest managers worldwide. Forests can provide an array of ecosystem services while helping to offset greenhouse gas emissions but are affected by natural and anthropogenic disturbances and climate change. These disturbances can interact with climate change to modify forest composition and productivity, thus affecting the capacity of forests to sustain services such as supplying wood assortments, maintaining biodiversity, and sequestering carbon [1]. Successful stand regeneration can mitigate these effects and is a keystone of sustainable forest management. While natural regeneration plays a substantial role in forest renewal—for example, about 44% of the area harvested in the past 20 years in Canada regenerated naturally [2]—the use of plantation forestry to regenerate specific species compositions offers added advantages for supporting sustainable forest management. Planted forests cover ~291 million ha worldwide, representing 7% of the world forests [3] and are expected to dominate future industrial wood supply [4].

The main reasons for controlling vegetation in boreal forests are to maintain conifer composition and to increase conifer yield. After harvesting, pure jack pine (*Pinus banksiana*

Lamb.) stands may be converted to mixed softwood or mixedwood conditions because of trembling aspen (*Populus tremuloides* Michx.) ingress [5]. Previous studies from across North America have shown that conifer release post-harvest leads to increases in conifer volumes from 49 to 5478% compared to untreated controls [6], while in Ontario, Canada, conifer release altered individual conifer growth between −49 and 556% compared with trees in untreated controls [7]. According to the Montréal Process sustainable forest management also includes maintaining biodiversity [8]. Trade-offs between biodiversity and productivity have been reported to occur in forest ecosystems. On average, a 10% loss in biodiversity leads to a 3% loss in productivity, although the most productive stands are typically monocultures [9]. Since vegetation management treatments are applied to enhance productivity through removal of competing, or undesirable, species, it seems reasonable to assume biodiversity will be reduced even as productivity increases.

The application of vegetation management treatments such as herbicides or brushsaws involved making trade-offs between criterion, such as stand-scale biodiversity and productivity [6,10,11]. While it is often expected that herbicides reduce biodiversity, e.g., [12], empirical studies have shown that herbicide applications do not typically result in species loss, but rather in shifts in abundances and evenness [13–15]. Tenth year post-treatment results have generally supported the hypothesis that forest vegetation release treatments do not produce tree monocultures [15]. However, under some conditions, stand-scale tree monocultures could result from multiple herbicide applications. Longer-term studies are needed to determine if these patterns persist in older plantations.

By properly managing stands, the potential exists to produce healthy trees with better stem form and wood quality to feed current and future industrial needs. Forests generally grow based on resources available to the individual trees. When resources are limited, tree growth slows, and different areas of growth are prioritized than when resources are plentiful. This simple idea also determines, in many cases, the wood properties we see in trees. For example, boreal forests with over 3000 established stems $ha^{-1}$ at the renewal stage typically produces good wood quality and stem form [16]. In contrast, a forest with less than 3000 established stems $ha^{-1}$ at the renewal stage produces less desirable wood properties and stem form, with this effect increasing as tree density decreases [17,18]. Stand density strongly influences wood properties such as wood density, number of growth rings per cm, timing of juvenile to mature wood transition [19], stem taper, stem branchiness, total height, diameter at breast height (dbh; 1.3 m), and wood mechanical properties [20]. Typically, as stand density decreases (i.e., below 3000 stems $ha^{-1}$), stem and wood properties worsen [16,21].

Best forest management practices must address competing demands, such as successful growth of crop trees following aerial herbicide release and increasing public concern about herbicide use in forestry. Thus, the objective of this study was to compare the effects of three release treatments (aerial herbicide spray, motor-manual brushsaw, complete competition removal via herbicides applied by backpack spray) and a control treatment (no vegetation control) on three individual criteria: [1] plant diversity, [2] stand-scale yield, and [3] individual crop tree responses at two jack pine sites. We hypothesized that long-term effects among the three release treatments on individual criteria would be similar, and that all release treatments would have a significantly positive effect on the three criteria compared with the control. We also hypothesized that temporary vegetation control would not lead to overstory monocultures at our study sites. We predicted that, in more intensive treatments (i.e., complete removal), individual crop tree survival and stem quality would increase, but plant diversity and branching would decrease. Furthermore, more intensive treatments would yield more conifers and less hardwood per ha than less intensive treatments. Here, we present data collected from two sites, established in 1992 and 1993, in northern Ontario, Canada, under the Vegetation Management Alternatives Program (VMAP; [22]). This research continues to address the lack of longer-term data needed by forest managers to implement more integrated vegetation management programs in boreal and sub-boreal forests.

## 2. Materials and Methods

### 2.1. Experimental Design

A field study was established in jack pine plantations in 1993 and 1994 on two sites to examine the effect of three release treatments and a control on individual crop tree growth, stand-scale growth and yield and plant diversity (Table 1). The first study site (Bending Lake) is located 53 km north of Atikokan, Ontario, Canada (48°57′ N, 92°02′ W) near the southwest shore of Clearwater West Lake. The second study site (E.B. Eddy) is about 120 km northwest of Sudbury, Ontario, in Olinyk (46°47′ N, 82°08′ W; 46°47′ N, 82°10′ W) and Oshell (46°46′ N, 82°04′ W) townships. The study was established as a randomized complete block design with four treatments (including a control) on each of seven blocks (Table S1). Four blocks were at the Bending Lake site and three blocks were at the E.B. Eddy site. The four treatments, in increasing order of intensity, were:

1.  No competition control (C), in which, aside from site preparation, post-logging vegetation was left undisturbed.
2.  Motor-manual brushsaw cutting (BS), with non-crop vegetation cut at ground level with a motor-manual brushsaw in June 1993 (Bending Lake) or with non-crop vegetation cut at 25 cm above groundline with a Husqvarna 165 clearing saw in October 1994 (E.B. Eddy).
3.  Operational single aerial herbicide application (AS), with 4 L ha$^{-1}$ of Vision® (1.5 hg acid equivalent [a.e.] ha$^{-1}$) applied in a total spray volume of 35 L ha$^{-1}$ using a G47T helicopter in late August 1992 (Bending Lake), or with 4 L ha$^{-1}$ of Vision® (356 g L$^{-1}$ a.e. glyphosate as a water soluble liquid) applied in a total spray volume of 34 L ha$^{-1}$ using a Bell 206 helicopter in late August 1993 (E.B. Eddy).
4.  Complete competition removal (complete removal; CR), with annual backpack spray applications of a 2% solution (400 mL glyphosate in 20 L water) of Vision® applied in August 1993, June 1995, and June 1996 (Bending Lake) or September 1993 and August 1994 (E.B. Eddy). The amount of Vision® sprayed via backpack application varied among plots depending on how much vegetation survived previous applications; however, comparing the efficacy of different Vision® rates was beyond the scope of this study. Each complete removal treatment plot was randomly located in a single aerial spray plot.

Complete competition removal in not typical of operational practices and is included as an extreme for the purpose of this study. Each treatment plot was a minimum of two ha at Bending Lake (for details, see [23]). At E.B. Eddy, each treatment plot was 50 × 100 m, except for herbicide application (100 × 200 m) and complete removal (40 × 40 m) plots, and berms covered about 5 to 10% of each plot area. In each replicate, treatments were randomly allocated to plots.

### 2.2. Vegetation Assessment

Before treatments were applied, 20 representative jack pine seedlings were systematically located within each assessment area. Each crop tree was identified with an aluminum tag attached to a numbered assessment pin at its base. At Bending Lake, selection was based on a 10 × 10 m grid of four trees by five trees. At E.B. Eddy, selection was based on an approximated 4 × 4 m grid of four trees by five trees with some trees closer or further apart [23]. Crop trees were recorded as dead, missing, healthy, or unhealthy. For each crop tree, other than the missing ones, total height, height to live crown, diameter at breast height (dbh), crown width, diameter of the five largest branches 50 cm above and below dbh, branching, and stem quality were assessed. After 25 growing seasons, biodiversity plots (10 × 10 m) were established off crop tree pins in a NW, NE, SW, or SE direction. Three of the 20 tree pins were chosen to provide maximum distance among plots. Vegetation was separated into seven layers (dominant tree cover, subdominant tree cover, 2–10 m trees and shrubs, 0.5–2 m trees and shrubs, <0.5 m trees and shrubs, herbs/grasses/sedges/ferns, and bryophytes/lichens) [13,24]. Biodiversity was identified to the species level (or genus

if species unknown) and percent cover was estimated. Three replicate growth and yield plots (11.28 m radius) were established off crop tree pins in each treatment plot, except for complete removal plots at E.B. Eddy (which only had two replicate plots due to limited space) to measure forest trees with a dbh ≥ 2.5 cm. To prevent overlap, a minimum 22.56 m radius was required between growth and yield plot centers. Growth and yield surveys were conducted in accordance with the Ontario Growth and Yield Standards as outlined in the Permanent Growth Plot and Permanent Sample Plot Reference Manual [25]. In total, data for 27 growth, yield, and biodiversity variables were collected at the end of the 25th growing season following initial treatment application: 18–22 July 2018 at Bending Lake site and 16–20 July 2019 at E.B. Eddy site. Further explanation of all variables assessed are provided in Table S2.

**Table 1.** Description of jack pine release study areas in northwestern Ontario, Canada.

| Site | Block | Geology | Preharvest Stand Conditions | Harvest | Site Preparation | Planting |
|---|---|---|---|---|---|---|
| Bending Lake | 1–4 (48°57′ N, 92°02′ W) | Flat to gently rolling; 1–10% bedrock outcrops; rapidly drained, coarse loamy to fine sand | V-17 [‡]: jack pine mixedwood/shrub rich forest | Clearcut; tree length, conventional cut and skid, 1986–1987 | Mechanical; heavy drags (barrels and chains), 1987 | Block 1: Jack pine; Cerkon shelter cone seeding, 3000 seedlings ha$^{-1}$ (1.8 × 1.8 m spacing), 1988 Blocks 2–4: Jack pine; overwintered Spencer-Lemaire 6 container stock; 3000 seedlings ha$^{-1}$ (1.8 × 1.8 m spacing), 1988 |
| E.B. Eddy | 1 (46°47′ N, 82°08′ W) | Glacial outwash plain; fine sands to silty loams (ES13.1) [*] | FRI 1974 [±]: Pw$_5$Pj$_3$Bw$_2$ Site Class 2; age 150; height 24 m; stocking 0.7 | Clearcut; tree-length, conventional cut and skid, 1988 [*] | Young's teeth, 1989 [*] | Jack pine; current 408 paper pot, planted with Potti Putki, June 27–29, 1989 [†] |
| | 2 (46°47′ N, 82°10′ W) | Glacial outwash plain; silty loam (ES15.1) [†] | FRI 1974[±]: Pj$_7$Sb$_2$Po$_1$ Site Class 2; age 85; height 20 m; stocking 0.8 | Clearcut; tree-length, conventional cut and skid, 1989 [†] | Young's teeth, 1989 [†] | Jack pine; overwintered FH408 paper pot, planted with Potti Putki, May 13–16, 1991 [†] |
| | 3 (46°46′ N, 82°04′ W) | Glacial outwash plain; silty, very fine sands to silty loams (ES15.1) [†] | FRI 1974[±]: Pj$_6$Sb$_2$Po$_2$ Site Class 2; age 90; height 21 m; stocking 0.6 | Clearcut; tree-length, conventional cut and skid, 1989 [†] | Young's teeth, 1989 [†] | Jack pine; current 165 Jiffy pot, planted with Potti Putki, June 14, 1991[†] |

[‡] [26]. [*] [27]. [±] Forest Resource Inventory stand description: Bw = white birch, Sb = black spruce, Pj = jack pine, Po = poplar, Pw = white pine. Superscripts refer to the proportion of total stocking (out of 10) occupied by a species. [†] [23].

### 2.3. Statistical Analyses

To assess differences in species diversity following release treatments, compositional indices (species richness (*S*), abundance (abun), Shannon entropy (*H'*), Simpson's dominance (*D*) and Heip evenness ($E_{Heip}$) were calculated using the methods described in [28]. Before calculating the indices, the seven canopy layers assessed in the field were condensed to four: L1 (dominant and subdominant tree cover), L2 (<0.5–10 m shrub layer), L3 (herbs, grasses, sedges and ferns), and L4 (bryophytes and lichens).

Diameter at breast height was measured for all live and dead trees, but height was recorded for only three to five trees per growth and yield plot at both sites. However, only live trees were used to calculate mean total height, quadratic mean diameter (QMD), dbh, height to live crown, branch diameter, rings per cm, inside and outside bark volume, basal area and density (number of trees), and merchantable volume (10 cm top diameter) in each plot. Merchantable volume of individual trees were calculated using Honer's conversion factors for jack pine [29]. Since height information is needed to estimate stem volume, heights of all live trees were estimated using a height-diameter model developed by [30]. Similarly, heights of hardwood trees were calculated using the models by [31]. Inside bark volumes of conifer trees were calculated using recently published volume equations [32,33] and inside bark volumes of hardwood trees were estimated using Honer's volume equations [34]. Total inside bark volumes, basal area, and number of trees from each plot were converted to per hectare values for both conifer and hardwood trees.

Statistical analyses were performed with R statistical software (Version 4.0.3, Vienna, Austria) [35] and the *lme4* function in the package *nmle*. Three-way analysis of variance (ANOVA), implemented through generalized linear mixed effects models (GLMM), was used to determine the effects of three release treatments and a control on (1) plant diversity characteristics, (2) stand-scale growth and yield characteristics, and (3) individual crop tree growth characteristics. For each analysis, treatment and site were fixed, and block was the random, independent variable. Residual analysis was performed to test for the assumptions of the three-way ANOVA. Normality was assessed using histograms and the Shapiro-Wilk's normality test and homogeneity of variances was assessed by Bartlett's Test. To meet assumptions of normality, the variables rings per cm, conifer density, and hardwood density were log transformed, and the variables gross total volume (GTV), basal area, merchantable volume, and individual jack pine volume were transformed using the cubed root function. When assumptions of normality were not met, we used nonparametric tests when appropriate. Poisson-distributed GLMMs were used to test for the effect of 'treatment + site + block' on species richness. Akaike Information Criterion corrected (AICc) score was used to select the most parsimonious model (using the R *MuMIn* package). *p*-values for normally distributed GLMMs were calculated using a type "II" mixed-model ANOVA, and *p*-values for Poisson-distributed GLMMs were calculated using the R *emmeans* package. Statistical significance was accepted at Bonferroni adjusted alpha levels of $p < 0.0125$ and $p < 0.0167$ for simple two-way interactions at Bending Lake and E.B. Eddy, respectively. Main effects analyses (pairwise comparisons) were determined using Tukey's honest significant difference (HSD) test with 95% confidence. Ordinal dependent variables (survival, stem quality, branching) were assessed using ANOVA with planned orthogonal contrasts [36]. Proportion of surviving stems was estimated to calculate individual jack pine survival. Ordinal dependent variables were transformed using the Anscombe transformation to meet assumptions of normality. The orthogonal contrasts compared were within treatments (CR vs. AS, CR vs. BS, AS vs. BS) and between control and treatments (C vs. (CR + AS + BS)/3).

The linear model for the ANOVA was:

$$Y_{ijk} = \mu + B_i + S + T + \mathring{a}_{ijk}, \tag{1}$$

where:
$Y_{ijk}$ is the calculated response from *i*th block, *j*th site, and *k*th treatment,
$\mu$ is the overall mean,
$B_i$ is the random effect of the *i*th block,
S is the fixed effect of the *j*th site,
T is the fixed effect of the *k*th treatment, and
$\mathring{a}_{ijk}$ is the pooled interaction effect of the *i*th block, the *j*th site, and the *k*th treatment with error term to test the treatment effect.

Statistical significance of differences among treatments was tested by pooling the interaction term with the error terms in the model. Characteristics for each individual criterion (plant diversity, non-crop tree growth and yield, and individual crop tree growth) were compiled as a percentage deviation from the baseline (ideal/maximum site conditions) (Table S2) and portrayed graphically in the form of radar diagrams. Radar diagrams are recommended as one way to handle multivariate data [37], and in this study the diagrams clearly illustrate differences among the effects of release treatments at the two sites.

## 3. Results

### 3.1. Plant Diversity

Plant diversity characteristics are displayed in Figure 1a. The effect of site was significant for softwood richness, tree abundance, herb abundance, and understory evenness (Table S3). Jack pine and trembling aspen as well as five other softwood tree species and four other hardwood tree species (>10 m) were present at both sites: balsam fir (*Abies balsamea* [L.] Miller.), white spruce (*Picea glauca* [Moench] Voss), black spruce (*Picea mariana*

[Miller] B.S.P.), red pine (*Pinus resinosa* Sol. ex Aiton), white pine (*Pinus strobus* L.), red maple (*Acer rubrum* L.), white birch (*Betula papyrifera* Marshall), largetooth aspen (*Populus grandidentata* Michx.), and pin cherry (*Prunus pensylvanica* L. f.) (Table S4). However, softwood richness was significantly higher at E.B. Eddy than at Bending Lake in all treatment and control plots, except complete removal. For example, average softwood richness by treatment ranged from 1.8 to 2.6 at Bending Lake compared with 3.0 to 4.2 at E.B. Eddy (Table 2). Hardwood richness was generally higher in all treatment plots than control plots at both sites, and tree abundance (% cover) was significantly higher in treatment plots at E.B. Eddy than in those at Bending Lake. Specifically, complete removal, aerial spray, and motor-manual brushsaw plots at E.B. Eddy had 11.1, 17.4, and 20% higher tree abundance, respectively, than their counterpart plots at Bending Lake.

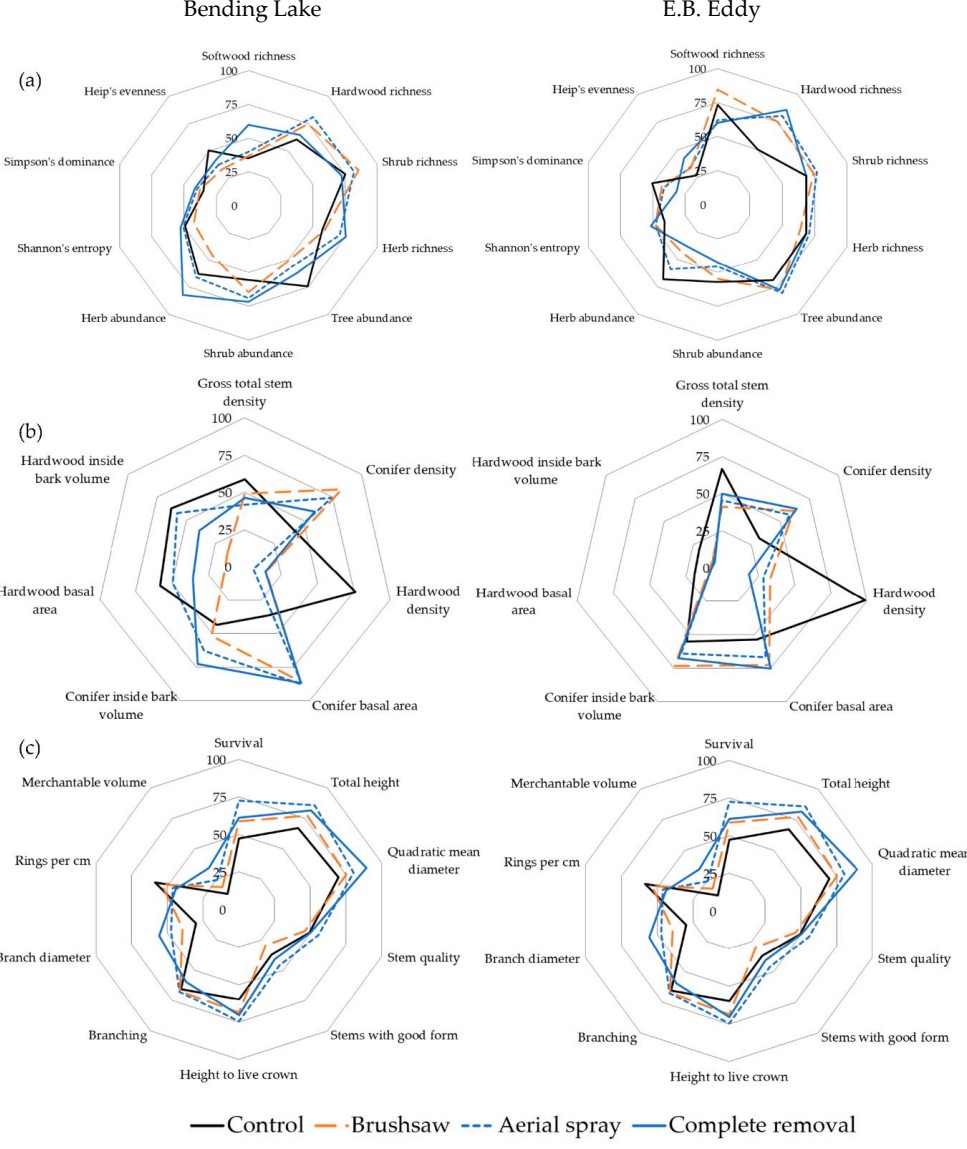

**Figure 1.** Radar diagrams proposed as a visual illustration of the effects of four release treatments on (**a**) plant biodiversity indicators, (**b**) stand-scale growth and yield indicators, and (**c**) individual crop tree growth indicators for two jack pine sites located in Ontario, Canada: Bending Lake and E.B. Eddy. Interior colored lines represent scoring for each criterion separated by treatment (aerial spray, motor-manual brushsaw, complete removal, and no competition control) 25 growing seasons after treatments applied. Values were calculated as percentage deviation from the baseline (ideal or maximum site conditions).

**Table 2.** Estimated marginal means (± SE) for biodiversity indicators (*n* = 85) grouped by site x treatment variables after vegetation management treatments applied on two sites (Bending Lake and E.B. Eddy) in Ontario, Canada. Different lowercase letters indicate statistically significant differences from Tukey's honest significant difference (HSD) test (pairwise comparisons between each treatment and control) in each site. Asterisk (*) indicates statistically significant results from pairwise comparisons between sites.

| Site | Treatment | Softwood Richness (Number of Species) | Hardwood Richness (Number of Species) | Shrub Layer Richness (Number of Species) | Herb Layer Richness (Number of Species) | Tree Abundance (Percent Cover) | Shrub Layer Abundance (Percent Cover) | Herb layer Abundance (Percent Cover) | Understory Diversity ($H'$) | Understory Dominance ($D$) | Understory Evenness ($E_{Heip}$) |
|---|---|---|---|---|---|---|---|---|---|---|---|
| Bending Lake | Control | 1.8 * (0.2) | 2.4 (0.3) | 9.0 (0.5) | 11.5a (0.6) | 52.0 (3.7) | 41.9 (5.0) | 47.2 (4.2) | 1.48 (0.18) | 0.34 (0.06) | 0.41 * (0.05) |
| | Brushsaw | 1.8 * (0.2) | 3.0 (0.3) | 10.3 (0.3) | 11.8a (0.6) | 35.4 (3.7) * | 48.8 (6.8) | 34.2 (4.8) | 1.29 (0.23) | 0.36 (0.08) | 0.31 (0.04) |
| | Aerial spray | 2.0 * (0.2) | 3.3 (0.2) | 9.8 (0.4) | 14.2b (0.8) | 39.0 (4.0) * | 52.2 * (7.1) | 49.5 (6.5) | 1.52 (0.22) | 0.38 (0.08) | 0.35 (0.07) |
| | Complete removal | 2.6 (0.3) | 3.0 (0.4) | 8.7 (0.5) | 15.1b (0.7) | 43.2 (4.0) * | 54.1 * (7.8) | 61.8 * (10.5) | 1.59 (0.22) | 0.40 (0.07) | 0.39 (0.05) |
| E.B. Eddy | Control | 3.7 (0.3) | 2.0 (0.4) | 8.2 (0.4) | 13.7 (0.5) | 48.3 (2.9) | 43.2 (4.3) | 51.1 (10.0) | 1.23 (0.28) | 0.48 (0.12) | 0.22 (0.04) |
| | Brushsaw | 4.2 (0.3) | 3.0 (0.2) | 8.9 (0.9) | 12.7 (1.3) | 54.4 (3.8) | 41.2 (5.6) | 33.0 (7.0) | 1.45 (0.30) | 0.41 (0.11) | 0.31 (0.05) |
| | Aerial spray | 3.1 (0.5) | 3.3 (0.4) | 9.2 (0.6) | 14.0 (0.8) | 56.4 (2.6) | 34.5 (6.4) | 44.2 (7.7) | 1.44 (0.29) | 0.40 (0.10) | 0.32 (0.05) |
| | Complete removal | 3.0 (0.4) | 4.0 (0.2) | 8.2 (0.9) | 13.7 (0.7) | 54.3 (3.8) | 32.1 (5.7) | 28.5 (3.8) | 1.56 (0.23) | 0.30 (0.08) | 0.39 (0.05) |

\* *p* values for fixed effects provided in Table S3.

Understory species richness was not significantly affected by vegetation management treatments and only one exotic species was observed (orange hawkweed, *Pilosella aurantiaca* (L.) F.W. Schultz & Sch. Bip.) in the herbicide treated plots at E.B. Eddy. However, no exotic species were observed at Bending Lake, and percent cover of the exotic species E.B. Eddy was low (0% in control and motor-manual brushsaw plots, 0–0.1% in aerial spray plots and 0–1% in complete removal plots). Aerial spray and complete removal plots at Bending Lake averaged 14 and 15 herbaceous species per plot, respectively, which was slightly higher than the 12 species found in both motor-manual brushsaw and control plots. Understory diversity and evenness were higher in treated than control plots at E.B. Eddy; however, differences were not significant. At Bending Lake, control plots had the highest understory evenness.

### 3.2. Stand-Scale Yield

Stand-scale yield characteristics are displayed in Figure 1b. Treatment effect was significant for all stand-scale characteristics (Table S5). Gross total stem density was highest in control plots at both sites, ranging from 3500 to 4000 stems ha$^{-1}$ (Table 3). Gross totals were higher at E.B. Eddy in all treatments except motor-manual brushsaw, which averaged 2910 stems ha$^{-1}$ at Bending Lake and 2331 stems ha$^{-1}$ at E.B. Eddy. All treatment plots had significantly higher conifer density, basal area, and GTVs than control plots at both sites. Motor-manual brushsaw plots had the highest number of conifers ha$^{-1}$ at both sites, with 2498 and 1842 trees ha$^{-1}$ at Bending Lake and E.B. Eddy, respectively. Between sites, Bending Lake plots (except complete removal) had more conifers ha$^{-1}$ and higher conifer basal area ha$^{-1}$ than those at E.B. Eddy, and were significantly higher for motor-manual brushsaw and aerial spray plots. At Bending Lake, motor-manual brushsaw and aerial spray plots had 656 and 500 more conifers ha$^{-1}$, respectively, compared with control plots that had 275. Opposing trends were evident for hardwood stand characteristics. Control plots had significantly higher hardwood densities, basal area, and inside bark volumes than treatment plots at both sites, and E.B. Eddy had the highest values for all three hardwood stand characteristics. For example, aerial spray, motor-manual brushsaw, control, and complete removal plots had 4.7×, 2.2×, 1.3×, and 1.2× more hardwoods ha$^{-1}$ at E.B. Eddy than at Bending Lake.

### 3.3. Individual Jack Pine Characteristics

Individual jack pine characteristics are displayed in Figure 1c. Controlling for site and block, treatment effect was significant for survival and branching, but not stem quality (Table 4). Effect of site was significant for survival, stem quality, and branching. Planned orthogonal contrasts revealed that treatment plots were associated with significantly higher survival rates compared with control plots. For example, jack pine survival rates in control plots were 47.5% and 36.7% at Bending Lake and E.B. Eddy, respectively, but ranged from 58.8–72.5% and 78.3–84.7% in treatment plots. However, control plots had significantly less branching compared with treatment plots, and higher treatment intensities (i.e., aerial spray, complete removal) were associated with the highest branching. Pairwise comparisons between sites revealed that trees in motor-manual brushsaw and complete removal plots at E.B. Eddy had significantly higher survival rates (26% and 16% higher, respectively) than those at Bending Lake (Table S6). Furthermore, stems were rated lower quality in all plots at Bending Lake by a difference ranging from 0.6 to 1.0 units (on a scale of 1 to 4) and there was significantly more branching in control plots at Bending Lake compared with E.B. Eddy.

**Table 3.** Estimated marginal means (± SE) for stand growth and yield indicators grouped by site x treatment combinations after vegetation management treatments applied on two sites (Bending Lake and E.B. Eddy) in Ontario, Canada. Different lowercase letters indicate statistically significant differences from Tukey's honest significant difference (HSD) test (pairwise comparisons between each treatment and control) within each site. Asterisk (*) indicates statistically significant differences from pairwise comparisons between sites.

| Site | Treatment | Gross Total Stem Density (Stems ha$^{-1}$) | Conifer Density (Stems ha$^{-1}$) | Hardwood Density (Stems ha$^{-1}$) | Conifer Basal Area (m$^2$ ha$^{-1}$) | Conifer Inside Bark Volume (m$^3$ ha$^{-1}$) | Hardwood Basal Area (m$^2$ ha$^{-1}$) | Hardwood Inside Bark Volume (m$^3$ ha$^{-1}$) |
|---|---|---|---|---|---|---|---|---|
| | | (*n* = 151) | (*n* = 81) | (*n* = 70) | (*n* = 81) | (*n* = 81) | (*n* = 70) | (*n* = 70) |
| Bending Lake | Control | 3518c (228) | 1233a (121) | 2285b * (287) | 11.8a (1.5) | 63.6a (8.1) | 10.6b * (1.3) | 57.0b * (6.9) |
| | Brushsaw | 2910bc * (183) | 2498c * (157) | 452a * (73) | 27.7b (0.7) | 148b (5.2) | 0.69a (0.2) | 3.53a (0.9) |
| | Aerial spray | 2402ab (177) | 2236bc * (145) | 180a * (53) | 28.4b * (0.8) | 160b * (4.7) | 0.50a (0.2) | 2.90a (1.2) |
| | Complete removal | 2045a * (102) | 1794b (71) | 430a (166) | 28.2b (0.8) | 154b (5.5) | 0.60a (0.2) | 3.27a (1.0) |
| E.B. Eddy | Control | 4011a (344) | 958a (163) | 3053b (353) | 10.8a (2.1) | 63.0a (10.9) | 13.8b (2.2) | 89.1b (19.0) |
| | Brushsaw | 2331b (224) | 1842b (114) | 989a (219) | 26.9b (0.9) | 154b (7.1) | 2.38a (0.7) | 13.6a (4.1) |
| | Aerial spray | 2561b (149) | 1736b (108) | 850a (143) | 22.7b (1.1) | 128b (4.9) | 1.65a (0.3) | 8.24a (1.5) |
| | Complete removal | 2725b (177) | 1925b (89) | 550a (173) | 27.3b (0.9) | 152b (8.8) | 1.31a (0.4) | 6.90a (2.3) |

* *p* values for fixed effects provided in Table S5.

**Table 4.** Analysis of variance results with orthogonal contrasts for individual jack pine tree survival ($n = 560$), stem quality ($n = 360$), and branching ($n = 360$) after vegetation management treatments (C = control, BS = motor-manual brushsaw, AS = aerial spray, CR = complete removal) applied on two sites (Bending Lake and E.B. Eddy) in Ontario, Canada.

| Source | df | Survival | | Stem Quality | | Branching | |
|---|---|---|---|---|---|---|---|
| | | F | *p*-Value | F | *p*-Value | F | *p*-Value |
| Site * | 1 | 5.8 | 0.016 | 30.41 | <0.001 | 4.12 | 0.048 |
| Treatment * | 3 | 17.71 | <0.001 | 1.14 | 0.334 | 8.67 | <0.001 |
| CR vs. AS | 1 | 1.651 | 0.099 | NA | NA | 2.39 | 0.017 |
| CR vs. BS | 1 | 1.068 | 0.286 | NA | NA | 3.31 | 0.001 |
| AS vs. BS | 1 | 1.79 | 0.074 | NA | NA | 0.78 | 0.435 |
| C vs. treatments | 1 | 7.06 | <0.001 | NA | NA | 3.8 | <0.001 |

* Means ($\pm$SE), separated by site and treatment, provided in Table S6.

Site and treatment significantly affected total height, rings per cm, and merchantable volume, and the interaction effects between site and treatment were significant (Table S7). Treatment also significantly affected quadratic mean diameter (QMD), height to live crown, and branch diameter, but the site x treatment interaction effects were not significant. Aerial spray plots had the tallest trees at both sites (12.7 m at Bending Lake and 12.9 m at E.B. Eddy) and all treatment plots had significantly taller trees ($p < 0.05$) than control (untreated) plots (Table 5). Between sites, trees at E.B Eddy were significantly taller in control (10.8 m) (p. adj < 0.001) and motor-manual brushsaw (12.4 m) (p. adj < 0.001) plots than those in the Bending Lake control (9.5 m) and motor-manual brushsaw (11.4 m) plots. Rings per cm were significantly higher for trees in control (1.7–2.2) than in treatment (1.3–2.0) plots at both sites, and trees at E.B. Eddy averaged 0.3–0.6 more rings per cm than those at Bending Lake. Merchantable volume was significantly higher in treatment than control plots, ranging from 0.04–0.07 m$^3$ tree$^{-1}$ in treatment plots compared with 0.03 m$^3$ tree$^{-1}$ in control plots at both sites. Quadratic mean diameter was significantly larger in treatment than control plots at Bending Lake; however, at E.B. Eddy it was slightly higher in control and motor-manual brushsaw plots than in aerial spray and complete removal plots. Height to live crown was largest in aerial spray plots at both sites and was significantly larger ($p < 0.05$) in all treatments than in control plots at Bending Lake, and in aerial spray and complete removal plots at E.B. Eddy. Branch diameter was significantly larger ($p < 0.05$) in all treatments compared with control plots at both sites, and aerial spray plots at Bending Lake had significantly larger branch diameters (p. adj < 0.001) than those in aerial spray plots at E.B. Eddy.

**Table 5.** Estimated marginal means ($\pm$SE) for individual jack pine growth characteristics ($n = 360$), grouped by site x treatment variables after vegetation management applied on two sites (Bending Lake and E.B. Eddy) in Ontario, Canada. Different lowercase letters indicate statistically significant differences from Tukey's honest significant difference (HSD) test (pairwise comparisons between each treatment and control) within each site. Asterisk (*) indicates statistically significant differences from pairwise comparisons between sites.

| Site | Treatment | Total Height (m) | Quadratic Mean Diameter (cm) | Height to Live Crown (m) | Branch Diameter (mm) | Rings (Per cm) | Merchantable Volume (m$^3$ ha$^{-1}$) |
|---|---|---|---|---|---|---|---|
| Bending Lake | Control | 9.5a * (0.4) | 11.0a (0.4) | 6.1a (1.1) | 11.5a (2.0) | 2.2b * (0.1) | 33.3a * (3.3) |
| | Brushsaw | 11.4b * (0.2) | 11.9b (0.3) | 7.2b (0.8) | 16.6b * (2.4) | 2.0ab * (0.1) | 97.4b * (6.1) |
| | Aerial spray | 12.7c (0.1) | 12.7b (0.3) | 7.8b (1.2) | 38.9d * (2.5) | 1.7a * (0.1) | 114.1c * (7.4) |
| | Complete removal | 11.9bc (0.2) | 14.2c (0.2) | 7.3b (1.1) | 22.8c (2.8) | 1.6a * (0.1) | 129.2d (5.1) |
| E.B. Eddy | Control | 10.8a (0.4) | 13.5ab (0.3) | 7.0a (0.2) | 13.3a (1.3) | 1.7b (0.1) | 32.6a (5.5) |
| | Brushsaw | 12.4b (0.2) | 13.4ab (0.5) | 7.0a (0.2) | 19.8b (0.7) | 1.4a (0.1) | 93.9b (5.8) |
| | Aerial spray | 12.9b (0.2) | 12.0a (0.3) | 7.6b (0.2) | 23.5c (0.9) | 1.2a (0.1) | 118.1c (7.4) |
| | Complete removal | 12.3b (0.3) | 12.9a (0.2) | 7.2ab (0.2) | 25.4c (1.0) | 1.3a (0.1) | 127.1c (4.8) |

* *p* values for fixed effects provided in Table S7.



## 4. Discussion

We analyzed long-term (25 year) individual crop tree growth and stand volume trade-offs between conifers and hardwoods in boreal and sub-boreal Canadian forests following three release treatments applied at two sites. Our results suggest that release treatments did not reduce plant diversity but benefitted stand yield and individual crop tree growth, with maximum benefits occurring in more intensive treatments.

### 4.1. Effect of Site and Species Ecology

Overall, site significantly influenced many of the diversity and productivity indicators. For example, tree species diversity and understory shrub and herb abundance and stand volumes were higher at E.B. Eddy than at Bending Lake; however, site-related differences must be interpreted cautiously. Although sampling intensity was identical, factors known to influence diversity and productivity differed between sites, for example, experimental plot size, site productivity and heterogeneity, and site preparation and planting timing. Relationships among these factors and species diversity are well known. Larger plots typically show higher species richness than smaller ones [38,39] due to the potential for higher heterogeneity in larger plots. Treatment plots at Bending Lake were minimally two times the area of those at E.B. Eddy; however, the sample area for each treatment was consistent at both sites. The higher diversity indices observed in some plots at E.B Eddy compared with Bending Lake are likely because the former is relatively more fertile based on its geological deposit and soil particle distribution [40]. While high levels of soil N usually decrease species richness (e.g., [41]), plant diversity generally increases from low fertility sites (here represented by the coarse loamy to fine sands at Bending Lake site) to intermediate fertility sites (here represented by very fine sands to silty loams at the E.B. Eddy site) based on a humped-back shaped relationship (e.g., [42]).

Silvicultural treatment differences, such as site preparation method, stand establishment density, and delays in temporary vegetation control, are confounded with site differences. Bending Lake and E.B. Eddy sites were trenched and bladed, respectively. As aspen roots are mainly contained within the top 12 cm of mineral soil [43], site preparation via trenching, which disturbs the mineral soil (not just exposes it, i.e., blading), may stimulate aspen regeneration through reducing the flow of sucker-inhibiting hormones within the roots, encouraging sucker formation [44]. Shade-intolerant hardwoods such as trembling aspen regenerate best in full light conditions [45,46] and moderately disturbed, fertile soils [47]. Both trenching and blading create a berm microtopography, but on our study site blading created 3 to 4 m wide berms that covered 5 to 10% of sample plot area. These wide berms were not planted with jack pine and were not treated during the brushsaw treatment; thus, they created nutrient rich refugia for hardwood and understory species in non-herbicide treated plots. Ref. [24,48] noted that untreated strips act as refugia. Jack pine was planted at 1.8 × 1.8 m at both sites, but due to ingress of naturals establishment densities were higher at Bending Lake. Stand density is known to influence branch habit [49], wood quality, and stand volumes [50,51]. Lastly, temporary vegetation control was delayed by two to four years at E.B. Eddy and by four to five years at Bending Lake (Table 1 and Table S1). Applying release treatments this long after harvesting allows establishment of understory vegetation such as grasses, forbs, and shrubs, which compete with shade-intolerant conifers such as jack pine, subjecting the latter to increased growth loss and mortality [23,52]. Thus, lower crop tree survival in all treatment plots at Bending Lake compared with E.B. Eddy likely resulted from the combination of low fertility and delayed temporary vegetation control at the site.

### 4.2. Treatment Efficacy

To fully understand the effects of vegetation management treatments on diversity and productivity, knowledge of treatment efficacy (i.e., cost and outcome) is needed. Based on a study of the cost effectiveness of controlling competitive vegetation such as aspen, [15] reported that herbicide treatments applied to white spruce plantations are three times more

cost effective than cutting treatments. Ref. [53] included 10th year Bending Lake and E.B. Eddy data in internal rates of return analyses and observed a 4.32% and 2.90% return for aerial herbicide and brushsaw treatments, respectively. Their findings also indicated that both brushsaw and herbicide treatments can be economically viable.

Brushsaw treatments are typically considered non-economical because at least two treatments [54,55] are often needed for success. Our results show that a single treatment application can be highly effective at reducing aspen densities; however, the cutting treatment at Bending Lake was more effective for controlling aspen than that at E.B. Eddy. We presume the difference in effectiveness can be attributed to differences in season, height of cut and site related differences. Research by [55] on aspen suggested that cutting in mid-summer at 50 cm, as applied at Bending Lake, is more effective for controlling aspen than cutting in early fall at 25 cm, as applied at E.B. Eddy.

Efficacy of glyphosate applications is known to be strongly influenced by seasonal timing and rate of application [56] and post-plant timing [52,57]. Although herbicide treatments were applied within the optimal seasonal time period and rates of application, delays in control of competitive vegetation may have significantly reduced the survival and growth of individual jack pine and other conifers. The critical period for jack pine, defined as the time during crop development when interspecific competition from weeds must be controlled to prevent significant yield losses, is two to three years after planting [6]. In our study, the period between planting and herbicide release treatments ranged from two to five years. Delaying temporary vegetation control by as much as two years after the critical period for jack pine may explain the low crop tree survival across all treatment plots. Applying herbicides within the critical period may increase crop tree survival and growth; however, early release may decrease the amount of understory plants such as blueberries [14].

*4.3. Treatment Effects on Diversity*

Our findings generally support our hypothesis that vegetation management, within the large intensity gradient we sampled, did not lead to overstory monocultures in boreal and sub-boreal forests. In fact, fifth year tree diversity data at E.B. Eddy predicts jack dominated stands would be established following all treatments [23]. However, we found increasing intensity of vegetation management appeared to increase species richness, particularly for hardwood species. This result may be due to the overall reduction in hardwood density in the treatment plots, which might have disrupted the typical dense vegetative reproduction of poplar and allowed other shade intolerant hardwoods time to establish before access to light became limiting. Despite efforts at complete removal in the most intensively treated plots, hardwoods persisted, which supports the hypothesis that tree monocultures do not develop after release treatments [15]. While multiple herbicide applications can reduce hardwood presence post-harvest, as applied in this study they did not remove them completely, probably because of their overall persistence in seedbanks and through sprouting as suggested by the direct regeneration hypothesis [58,59]. The fact that other vegetation components remained in these areas after 25 years suggests that aerial spray and complete removal treatments may offer options for alternative management outcomes. However, verification that these species persist through continued stand development requires further investigation.

Intensification of vegetation management treatments also had positive, or non- significant, effects on shrub and herb layer richness, which was unexpected. This finding may be the result of increases in light availability for the understory through decreases in hardwood proportion in treatment plots [14,60], an increase in substrate diversity from repeated minor disturbance and removal of smaller trees, or the increased overstory diversity, all of which increase vascular plant species richness and shrub species richness in boreal forests [61,62]. Indeed, high stand density, and resulting reduced light, can override the generally positive effects of mixed forests on understory diversity [63]. Non-significant effects of intensification on evenness suggest that while intensification improved species

richness of the understory, it did not cause only few species to dominate. Understory vegetation cover in aerial spray and complete removal plots included understory plants such as wild blueberries (*Vaccinium angustifolium* Aiton) and (*Vaccinium myrtilloides* Michx.), sweet fern (*Comptonia peregrina* [L.] J.M. Coult.), beaked hazel (*Corylus cornuta* Marshall), wild red raspberry (*Rubus idaeus* L.), bush honeysuckle (*Diervilla lonicera* Miller), *Alnus* spp., and *Salix* spp. These plant species are vital wildlife food sources and support previous research that suggested herbicide treatments maintain wildlife food source availability [23]. Although these results suggest that most understory plant species in mid-successional boreal and sub-boreal forests can withstand intensive silvicultural management regimes, public acceptance of herbicides is low [64]. If herbicide use is not permitted in the management area, or if a chemical-free alternative is preferred, motor-manual brushsaw may offer options for managing mixedwood stands [23]. In either case, vegetation control to maximize planted crop tree growth is not incompatible with maintaining diverse understory communities [65]. However, future studies should consider if or how the understory community composition is altered by release treatments.

The intermediate disturbance hypothesis suggests that diversity should be highest when there are moderate levels of disturbance [66]. However, 25 years after harvesting at our study sites, understory species diversity mostly increased with increasing management intensity. In the first three years where herbicides were ground applied annually in complete removal plots, deciduous tree and tall shrub species abundance was lowest in complete removal plots [23], but quickly rebounded, as glyphosate has no effect on plants germinating post-treatment. If glyphosate application in complete removal plots had continued, these plots would likely have lower species diversity, as only highly tolerant species or good colonizers would persist [67]. Furthermore, multiple studies provide evidence that anthropogenic disturbances such as harvesting and silviculture can facilitate invasions by exotic plant species in boreal forests [13,14,68]. Shade-intolerant exotic species, such as common dandelion (*Taraxacum ceratophorum* [Ledeb.] DC.), colonize the open spaces created immediately after disturbance [14]. Such species, however, were generally not recorded in the closed canopy conditions observed 25 years post-harvest.

*4.4. Treatment Effects on Stand-Scale Yield*

Our findings generally support our hypothesis that more intensive treatments would yield higher conifer volume and less hardwood volume than less intensive treatments. Jack pine density, basal area, and stand volume were all significantly higher in treatment plots compared with the control; however, volumes were higher for single applications of glyphosate than the more intensive multiple applications. Why the complete removal treatment did not produce maximum GTV as predicted is unknown; however, the most probable explanation is that jack pine simply allocated excessive amounts of energy to branch growth. Removal of competition at ground level, combined with low stand densities, likely allowed trees in complete removal plots to allocate more wood to branches to support more foliage. This explanation is supported when comparing crown width values among treatment plots. Complete removal plots had the largest crown width (4.6 m) relative to other treatment plots (3.1 m) and control plots (2.6 m) (unpublished data).

Our findings suggest that jack pine volumes could be increased through vegetation management, but there may considerable opportunities to further improve the yield of these jack pine plantations, possibly through a combination of vegetation management and tree improvement. The mean annual volume increment (MAI) for treated jack pine plantations in this study was, on average, 6.3 m$^3$ ha$^{-1}$ in aerial spray plots and 2.6 m$^3$ ha$^{-1}$ in control plots. While these numbers clearly show a significant beneficial role of vegetation management, our highest values are still far lower than the MAI of 10.0 m$^3$ ha$^{-1}$ reported for the best jack pine provenances growing in Ontario [69]. Our findings also suggest that current forest management models, such as Forest Vegetation Simulator (FVS) [70], may underestimate volume yields. Using 10th-year post-treatment data from the Bending Lake and E.B. Eddy sites, [53] applied FVS to predict GTV to an arbitrary age of 70 years. For the

period of this study (i.e., 25 years post-treatment), they predicted that GTV for the untreated control and complete removal treatments would yield the lowest and highest volumes, respectively, and that the GTV of these stands would be between 50 and 100 m$^3$ ha$^{-1}$. These predictions were well under actual values. For example, in the Bending Lake study, the GTVs ranged from 75 m$^3$ ha$^{-1}$ for the control to 161 m$^3$ ha$^{-1}$ for the aerial spray. The volume gains associated with the complete removal treatment at year ten post-treatment did not persist as predicted. We presume that underestimation of GTV is attributed to model functions within FVS. This is important, because managed jack pine stands contribute about 600 million m$^3$ GTV [71] to the 720 million m$^3$ GTV of jack pine growing in Ontario [72]. Despite the contribution of jack pine plantations to Ontario's wood supply, very few studies have reported on the long-term (i.e., greater than 20 year) growth of jack pine plantations [73]; thus, this study helps to fill in an information gap.

In this study, we calculated volumes (GTVs) using real data (i.e., height, diameter, etc., measured in the plots) by applying recently published volume equations developed for plantations. Therefore, GTVs used this analysis are more accurate than those projected using FVS. The volume models used in the FVS calibrated for Ontario incorporated models developed for natural stands and volumes of trees grown in plantations differ from those in natural stands [32,33]. Ref. [71] suggested that the use of tools to estimate the production potential of managed stands may significantly underestimate their productivity. Ref. [73] confirmed this bias for a study of jack pine plantations near Nipigon, Ontario. They reported a mean GTV of 307 m$^3$ ha$^{-1}$ (range of 2–368 m$^3$ ha$^{-1}$).

In addition, volumes may be enhanced through complementarity. Recent research on plant species complementarity suggests that productivity of conifer stands may be improved through mixed species plantations [74]. The possibility mixed species forest stand productivity might differ from that of pure stands has long intrigued growth and yield researchers [75]. Ecological theory suggests that when two or more species use resources differently, the resources may be more completely exploited by plants growing in mixture than by those growing as single species, leading to higher productivity in mixed species stands [74,76]. However, recent research has highlighted that complementarity can only occur when species do not overlap in resource acquisition. For example, contrasting shade tolerance levels among constituent species are needed to observe higher productivity with increasing diversity globally [77], and in the boreal forest specifically [78]. This discrepancy may be why stand-scale volumes were higher in near pure stands of jack pine than in intimate mixtures of aspen and jack pine, given that both are intolerant pioneers. Though these stands are relatively young and diversity effects have been frequently observed to increase through time [59,79–81] it is unlikely that trees in the control plots (i.e., plots with higher proportions of mixture) would be able to outgrow the near pure jack pine stands in an operational timeframe. Additionally, these stands have reached canopy closure, at which point positive diversity effects become most evident [81]. Future studies should consider mixtures of species that complement jack pine in shade tolerance [78], rooting structure [82], or nitrogen fixation [83]. We found no evidence that mixtures of aspen and pine increase tree growth. Increased productivity in mixtures has not been consistently demonstrated by analyses of the effect of composition on stand growth [75,84].

### 4.5. Treatment Effects on Individual Jack Pine Characteristics
#### 4.5.1. Survival, Height, and Diameter

Jack pine survival, height, and quadratic mean diameter were all significantly greater in treatment compared to control plots. These findings are generally in line with our hypotheses as well as previous studies that have shown that jack pine survival, height growth, and diameter can be strongly influenced by competition with trembling aspen and other woody species. Typically, jack pine height growth and diameter are more sensitive to competition than survival [7,52,55]. This hierarchical pattern of competition affecting conifers has been well documented in the literature [85]. However, in our study we found a substantial positive effect of treatment on jack pine survival, particularly at E.B. Eddy.

We attribute the relatively low survival rate of jack pine in the controls at the E.B. Eddy site to severe competition from overtopping aspen. While water and nutrients can be limiting for natural jack pine stands [86], after disturbance, light is a limiting resource for individual jack pine [87]. Furthermore, competition for light is asymmetrical, meaning that larger trees obtain disproportionate access to light for their size [88]. After clearcuts without vegetation management, intolerant hardwoods such as trembling aspen can easily recruit at high densities and can outgrow planted jack pine [5]. Once jack pine seedlings are overtopped, it becomes increasingly difficult for them to access light, reducing survival and height and diameter growth. Our results reinforce the assertion that if forest managers aim to regenerate a high proportion of jack pine, vegetation management is essential.

Our findings are comparable to those of [6], who reported gains in wood volume of 49–5478% in northern forests. The results of the single unpublished jack pine study reported in that review showed wood yield gains at 10 years post treatment of 116%, compared to those found in our study, which were wood gains at 25 years post treatment of 288–390%. Ref. [57] have since confirmed that survival, height growth, and diameter of jack pine are increased by herbaceous vegetation control. Continued monitoring of these sites through rotation is essential to determine whether increases in merchantable volume are maintained, or amplified, as stand development occurs.

Lack of significant difference in conifer response to herbicide and brushsaw treatments has been previously reported [7,89]. We attributed the lack of differences in this study to the fact that both the herbicide and mechanical cutting treatments were applied in the proper season and post-treatment jack pine densities were relatively high. Stand density is known to affect individual jack pine survival and growth. When natural pure (or nearly pure) jack pine stands are too dense, survival and height may be reduced due to high intraspecific competition for light and nutrients [90]. Low stand densities can also decrease jack pine height [91], because individuals will allocate more mass to branches when they are not competing as intensely for light [92]. This increased allocation to branches leads to a deterioration in stem form [69]. At optimal stand densities, lower branches on the stem decrease in growth and/or die from shading and competition [21]. Consequently, more mass may be allocated to the stem, increasing diameter. Stem diameter is a major wood quality attribute for the sawmilling industry. With increasing stem diameter, logging and manufacturing costs decrease, whereas lumber volume recovery and grade yield increase significantly.

Despite observed increases in height and quadratic mean diameter in our treatment plots, stand densities were below those recommended (3000 stems ha$^{-1}$) to maintain jack pine quality wood [17,21], suggesting further room for improvement. Planting density (1.8 × 1.8 m) at our sites was within spacing recommended to maximize jack pine growth and productivity. However, this initial spacing provides the desired stem density of 3000 stems ha$^{-1}$ when most seedlings survive. Clearly, control plots did not meet this density since jack pine mortality rates were high, possibly due to ingress of aspen and subsequent interspecific competition for resources, reaffirming that site preparation and planting alone cannot regenerate near-pure jack pine stands. Treated plots also did not achieve optimal density, indicating that higher planting densities, tree improvement focused on increasing seedling survival, pre-harvest treatment, or further silvicultural interventions (such as thinning of undesirable species) need to be investigated.

### 4.5.2. Stem Form, Knot Size, and Branch Diameter

Silviculture activities in jack pine forests have profound effects on how the trees grow and the resulting wood properties. Jack pine is a species that typically has many thousands to tens of thousands of trees on a site after a disturbance, with fire being the main natural disturbance affecting this species. In a study by [93], increasing tree spacing increased jack pine crown length, live crown ratio, and crown width. Additionally, branches were larger in the widest spacing as were stem taper, sweep, and wobbling. This finding provides evidence that jack pine is not a species that benefits from excess space. All these factors

lead to less desirable wood for lumber production. In a study by [20] increased black spruce spacing resulted in similar trends for stem features as those found for jack pine, i.e., decreased lumber grades and reduced yield of high-quality lumber. The main downgrades were due to knots and wane, which result from larger knots and more taper in widely spaced trees. They also found that in the lowest stand density, the highest grades of lumber were not found in any of the trees, meaning the value of each tree was significantly lower. In both the [93] and [20] studies, the lowest stand density produced the lowest quality lumber and wood quality was not comparable to that of trees from natural stands.

In many Canadian provinces, jack pine is an important commercial species and represents a substantial component of the total softwood harvest. The wood is used in the lumber and pulp and paper industries, with lower logs going to sawmills and upper logs to pulp mills. The lumber is strong and rated for structural building, while the fiber has enough length for quality paper. Much of the strength properties of jack pine are based on its wood properties and, as discussed above, stand density affects both wood mechanical properties and density [20]. As stand density decreases more wood is produced in each tree, but the mechanical strength of the wood decreases. This finding is significant and requires a more detailed study of wood properties at different stand densities to grasp the full effect of stand density management. Ref. [53] also found that stand density management increased jack pine attributes such as height and diameter, GTV, and projected value of fiber relative to controls.

Although stem quality was unaffected by vegetation management, average stem form was worse at our sites than in similar studies reporting on extrinsic tree quality. For example, sixteen years after herbicide and cutting treatment on a white spruce plantation in northwestern Ontario, 80% of trees in all treatment plots were classified as having good stem form [36] compared with 57, 58, and 62% of living trees in complete removal, motor-manual brushsaw, and aerial spray plots, respectively, after twenty-five years. This result may be due to low stand densities as previously mentioned. Based on a jack pine density management diagram for boreal Ontario [18,94], forest managers should aim for stand densities of at least 3000 stems ha$^{-1}$ to ensure early crown closure, which is required to minimize branchiness [90]. Therefore, increasing initial planting densities (above the standard $1.8 \times 1.8$ m spacing that yields 3084 stems ha$^{-1}$) to account for tree mortality may allow forest managers to optimize stand merchantable volume. Another reason for reduced stem quality and increased branching in treatment plots in our study may be the genetics of our planted crop trees. Jack pine does not usually grow perfectly straight, resulting in irregular shapes, and is usually knotty. However, certain characteristics of jack pine (i.e., growth, branching habit) can be genetically selected for to get better quality wood and increases in overall wood volume [95].

Information about the effects of stand density on wood quality is important to move management towards maximizing growth while maintaining wood quality. Research is underway to look at genetic trials for growth combined with wood quality research to link the growth rates and tree properties to the internal wood properties as a means of selecting families that show increased growth rates while maintaining wood quality.

## 5. Conclusions and Recommendations

This study offers one of the longest-term evaluations of operational vegetation management options for releasing sub-boreal and boreal jack pine plantations (aerial herbicides, motor-manual cutting), and it includes extremes (untreated control, complete removal) as benchmarks. Moreover, it provides information about treatment effects on plant diversity, stand-scale productivity, and tree-scale growth and quality, in interaction with inherent site characteristics. Our results suggest that plant diversity and productivity in boreal jack pine forests are significantly influenced by vegetation management treatments; however, none of the treatments maximized all 27 indicators studied; therefore, forest managers selecting treatments are faced with trade-offs. With the increase in mass timber production, we must maintain forests that will meet the grades for existing and new products and provide a

constant sustainably managed supply to mills. Future studies should investigate the longer term (ideally through at least one rotation) effects of vegetation management treatments on higher jack pine densities (whether through increased initial planting densities, more vigorous thinning regimes, or infill planting).

Research on the criteria and indicators for the conservation and management of boreal forests not discussed here is also needed to make informed policy decisions and progress toward sustainable forest management. These assessments should include forest ecosystem health, soil and water resource conservation, and the maintenance of socio-economic benefits [8]. Given our observed trade-offs among vegetation management treatments for plant diversity, stand-scale growth and yield, and crop tree growth and form, it is highly likely that these treatments have trade-off implications for other ecosystem services such as nutrient cycling and wildlife diversity [96–98]. As well, these treatments have socio-economic implications. Investigation of societal and economic values associated with vegetation management alternatives should focus on the maintenance of the "licence to operate" [99–102]. Coupled with this, recent research suggests that First Nations opposition to herbicide use is complex and multifaceted [103]. Further research is needed to understand these complexities, through collaborative work that meaningfully engages First Nations communities concerned about forest management and that weaves Indigenous ecological knowledge into projects informing vegetation management programs. In addition, research on intrinsic values such as wood density (ring density, earlywood ring density, and latewood ring density), percentage of latewood in the ring, and wood bending properties [104] may also be prudent to support both traditional (i.e., sawmilling, pulp and paper) and emerging (i.e., bioenergy, biochemical extractables) forest sectors.

**Supplementary Materials:** The following are available online at https://www.mdpi.com/1999-4 907/12/3/370/s1, Table S1: Summary of jack pine release treatment details applied on two sites in Ontario, Canada, Table S2: Data collected on crop trees, noncrop vegetation growth and yield, and biodiversity during the 25th growing season after aerial herbicide application, manual brushsaw cutting, and complete competition removal at two sites (Bending Lake and E.B. Eddy) in Ontario, Canada [105–107], Table S3: *p* values from three-way analysis of variances (ANOVAs) for biodiversity indicators (*n* = 85) after vegetation management treatments applied on two sites (Bending Lake and E.B. Eddy) in Ontario, Canada. Statistical significance was accepted at Bonferroni adjusted alpha levels of *p* < 0.0125 and *p* < 0.0167 for simple two-way interactions at Bending Lake and E.B. Eddy, respectively, Table S4: Species list for Bending Lake and E.B. Eddy sites 25 growing seasons after jack pine release treatments. Bolded names refer to species found only at the Bending Lake site and those with asterisk (*) were found only at the E.B. Eddy site, Table S5: *p* values from three-way analysis of variances (ANOVAs) for stand growth and yield indicators (*n* = 82) after vegetation management treatments applied on two sites (Bending Lake and E.B. Eddy) in Ontario, Canada. Statistical significance was accepted at Bonferroni adjusted alpha levels of *p* < 0.0125 and *p* < 0.0167 for simple two-way interactions at Bending Lake and E.B. Eddy, respectively, Table S6: Adjusted means for individual jack pine survival (*n* = 560), stem quality (*n* = 360), and branching (*n* = 360), grouped by site x treatment variables after tending treatments applied on two sites (Bending Lake and E.B. Eddy) in Ontario, Canada. Statistical significance at *p* < 0.05, calculated only for variables with a significant main effect of site. Stem and branching were rated on a scale of 1 to 4, with 1 having best form and least branching, Table S7: *p* values from three-way analysis of variances (ANOVAs) for individual jack pine growth indicators (*n* = 360) after vegetation management treatments applied on two sites (Bending Lake and E.B. Eddy) located in Ontario, Canada. Statistical significance was accepted at Bonferroni adjusted alpha levels of *p* < 0.0125 and *p* < 0.0167 for simple two-way interactions at Bending Lake and E.B. Eddy, respectively.

**Author Contributions:** Conceptualization, H.D.D. and F.W.B.; methodology, H.D.D., F.W.B., E.B.S., M.L., M.S. and J.D.; software, H.D.D.; validation, N.T., E.B.S. and M.L.; formal analyses, H.D.D. and M.S.; investigation, F.W.B. and J.D.; resources, F.W.B. and J.D.; data curation, J.D.; writing—original draft preparation, H.D.D., F.W.B., N.T., E.B.S., M.L. and M.S.; writing—review and editing, J.D.; visualization, H.D.D. and F.W.B.; supervision, F.W.B.; project administration, F.W.B. and J.D.; funding acquisition, F.W.B. and J.D. All authors have read and agreed to the published version of the manuscript.

**Funding:** Data collections for this study were funded by the Ontario Ministry of Natural Resource and Forestry.

**Data Availability Statement:** The data presented in this study are available on request from the corresponding author.

**Acknowledgments:** The authors acknowledge Bob Wagner for initiating and administering the Vegetation Management Alternatives Program (VMAP), without which the Bending Lake and E.B. study sites would not have been conceived. Trial establishments and treatment installations were administered by Chris Holstedt and Andrée Morneault at Bending Lake and E.B. Eddy, respectively. The authors also thank Justin Viljakainen, Cole Barban, Amy Bolduc, Madison McCaig, Elaine Mallory, Erika Mihell, Christine St. Jules, Ana Kostic, Kayla Hayden, Riley Stobie, and Sierra Rutland for their help in collecting the twenty-fifth-year measurements. Thanks also to Lisa Buse (OFRI) and Mike Brienesse (OMNRF) for their thoughtful reviews of the draft manuscript.

**Conflicts of Interest:** The following authors, H.D.D., N.T., M.L., M.S., J.D., and E.S. declare no conflict of interest. F.W.B., an employee of the funder, had a role in the design of the study; in the collection, analyses, and interpretation of data; in the writing of the manuscript, and in the decision to publish the results.

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
