# Peer review of "Trade-Offs among Release Treatments in Jack Pine Plantations: Twenty-Five Year Responses"

_forests, doi:10.3390/f12030370_

Round 1
Reviewer 1 Report
Review of the manuscript “Trade-offs among release treatment in jack pine plantations: Twenty-five year responses”.
General comments:
The study is extremely important because it clearly shows that certain suppositions taken as dogma are not true in all conditions. Intensive forestry oriented to the production of wood assortments needn’t lead to biodiversity loss. The most important message of this study is that reasonable forest management must be adopted based on investigation of local condition and general rules for forest management in global scale do not exist.
In the study I found small errors but they do not affect the whole impression (see specific comments). The study is very good.
Specific comments:
Lines 41. Jack pine wood production includes not only wood fibre, so the recommendation is the term wood fibre should be replaced wood assortments.
Lines 75 and 80. The number of stems per ha recommended for the optimal growth and quality is introduced as 3000. However, according to this paragraph it is not clear, in which age of stand. During reading the text I found out that this figure showed the number of seedlings. But this fact should be displayed in this paragraph, too.
Line 85. The method complete competition removal should be extended, for instance complete competition removal by herbicides applied by backpack spray. The reason is that actually two methods are very similar, herbicide use, and in this paragraph such similarity is not clear.
Line 89-90. The formulation of hypothesis should be more precise. The study deals with temporary vegetation control and such control is also discussed according to the results. In discussion, there is a note that permanent control could lead to other results, but authors should keep the aim of the study. So the recommendation is that the term vegetation control should be extended to temporary vegetation control.
Table 1. The abbreviation Sb written in the column Preharvest stand condition is not explained in the legend (Line 158).
Line 164. The citation (Bell et al.) does not respect the citation rules.
Line 279. Criteria for stem quality are missing. Which stems are good, worse and the worst?
Page 15 (Lines 323-365). Why numbers of cited authors are written in italics?
Lines 455-457. Authors should discuss the results shown in the study. The crown width is not mentioned in the chapter Results. So the sentences dealing with the crown width should be removed.
Lines 942-949. These items of chapter References (items 104-107) are missing in the manuscript.
Author Response
General Response to Reviewer Comments
We are extremely appreciative of the enthusiastic comments and helpful suggestions of the reviewers. All suggestions to improve the manuscript have been considered, and specific replied to the individual comments by reviewers are included below.
General comments:
The study is extremely important because it clearly shows that certain suppositions taken as dogma are not true in all conditions. Intensive forestry oriented to the production of wood assortments needn’t lead to biodiversity loss. The most important message of this study is that reasonable forest management must be adopted based on investigation of local condition and general rules for forest management in global scale do not exist.
In the study I found small errors but they do not affect the whole impression (see specific comments). The study is very good.
Reply: We thank the reviewer for their thorough comments about our manuscript. Concerns regarding specific comments have been responded to below.
Specific comments:
Lines 41. Jack pine wood production includes not only wood fibre, so the recommendation is the term wood fibre should be replaced wood assortments.
Reply: Thank you for this suggestion. This specification has been done.
Lines 75 and 80. The number of stems per ha recommended for the optimal growth and quality is introduced as 3000. However, according to this paragraph it is not clear, in which age of stand. During reading the text I found out that this figure showed the number of seedlings. But this fact should be displayed in this paragraph, too.
Reply: The 3000 stems per hectare threshold refers to number of stems required at canopy closure. We have rectified this in Lines 76-78.
Line 85. The method complete competition removal should be extended, for instance complete competition removal by herbicides applied by backpack spray. The reason is that actually two methods are very similar, herbicide use, and in this paragraph such similarity is not clear.
Reply: We thank the reviewer for adding this suggestion regarding clarity of our objectives. In the paragraph mentioned, we now define aerial spray as “aerial herbicide spray” and complete competition removal as “complete competition removal via herbicides applied by backpack spray” (lines 87-88).
Line 89-90. The formulation of hypothesis should be more precise. The study deals with temporary vegetation control and such control is also discussed according to the results. In discussion, there is a note that permanent control could lead to other results, but authors should keep the aim of the study. So the recommendation is that the term vegetation control should be extended to temporary vegetation control.
Reply: We thank the reviewer for noticing the improper use of the term “vegetation control” where we mean “temporary vegetation control”. Throughout the document, where appropriate, we have changed the term “vegetation control” to “temporary vegetation control”.
Table 1. The abbreviation Sb written in the column Preharvest stand condition is not explained in the legend (Line 158).
Reply: We thank the reviewer for noticing this oversight. The abbreviation Sb is now explained in the legend (line 162).
Line 164. The citation (Bell et al.) does not respect the citation rules.
Reply: This specification has been done.
Line 279. Criteria for stem quality are missing. Which stems are good, worse and the worst?
Reply: Stem quality criteria is described in Table S2 in the Supplementary Materials. We have now added a note about the scaling in the results for easier comprehension when reading the main text (lines 289-290).
Page 15 (Lines 323-365). Why numbers of cited authors are written in italics?
Reply: Thank you, the italics have been removed.
Lines 455-457. Authors should discuss the results shown in the study. The crown width is not mentioned in the chapter Results. So the sentences dealing with the crown width should be removed.
Reply: We thank the reviewer for this suggestion. Lines 456-458 regarding crown width have been removed.
Lines 942-949. These items of chapter References (items 104-107) are missing in the manuscript.
Reply: These items are referred to in the supplementary materials only. According to the journal specifications, we thought we were to include them in the references. We will gladly follow any instructions provided by the Editorial office in that regard.
Reviewer 2 Report
Dear Authors,
I had a pleasure to review your manuscript entitled “Trade-offs among release treatments in jack pine plantations: Twenty-five year responses”.
I find the reviewed paper as a very interesting and timely.
There are no objections to the reviewed work. The manuscript was prepared very carefully and extensive empirical materials were properly compiled in terms of statistics. The results were presented in a synthetic way and fully discussed.
Although Canadian forestry has its own specificities, I fully believe that the problems raised in your work will be also of interest to foresters from different regions worldwide.
Yours sincerely,
Reviewer
Author Response
General Response to Reviewer Comments
We are extremely appreciative of the enthusiastic comments and helpful suggestions of the reviewers. All suggestions to improve the manuscript have been considered, and specific replied to the individual comments by reviewers are included below
Reviewer 2
Dear Authors,
I had a pleasure to review your manuscript entitled “Trade-offs among release treatments in jack pine plantations: Twenty-five year responses”.
I find the reviewed paper as a very interesting and timely.
There are no objections to the reviewed work. The manuscript was prepared very carefully and extensive empirical materials were properly compiled in terms of statistics. The results were presented in a synthetic way and fully discussed.
Although Canadian forestry has its own specificities, I fully believe that the problems raised in your work will be also of interest to foresters from different regions worldwide.
Yours sincerely,
Reviewer
Reply: We are thankful to the reviewer for taking their time to read our manuscript. We are happy they enjoyed our work and hope we can only improve upon the manuscript based on edits from the other reviewers’ comments.
Reviewer 3 Report
The revised manuscript represents a long-term study of forest plantations focused on productivity and biodiversity. I appreciate the long-term observation that brings valuable results in assessing stand productivity and individual tree characteristics under different treatments. However, the weak point is the assessment of diversity indices. Since diversity observations were conducted 25 growing seasons after the treatments, I doubt the interpretation of results. The authors stated that treatments have a non-significant impact on diversity indices. In contrast, the impact on stand productivity was positive, which could be interpreted as all of the treatments are suitable management options even in the case of biodiversity protection. Since diversity sampling was not continuous, we do not know from the study the effect of treatments on diversity in few years after treatment. It is presumed that the effect of the treatment was very negative and could last for years. It is more likely that a non-significant impact on diversity was obtained due to vegetation's regeneration ability even after intensive treatment. Twenty-five years is enough time for vegetation to regenerate, so the results did not show that treatments have no impact on diversity. It only shows that after 25 years, vegetation can mitigate treatments' effect, but we have no information about processes between these years. I agree with the authors that trade-offs are needed in decision-making when forest management activities are planned. Ecosystem services have to be carefully assessed to quantify if more significant volumes after 25 years are worth 5, 10 maybe 15 years of disrupted forest ecosystem functions.
The results of the study are valuable and interesting from the forest production point of view. However, diversity measures have to be improved and carefully discussed.
Specific comments:
Introduction
Some basic descriptions of the vegetation communities' character that occur naturally in the area are missing, same as characteristic of jack pine plantations – what are the most widespread shrub species that use to be removed? How diverse these communities use to be?
Methods
A map of the study area with the location of plots could be added.
As stated in Methods, the study sites have different areas. Did the authors try to add site area as a random variable in models to test the influence?
Diversity was assessed as the number and abundance of species and standard diversity indices. To better insight into treatment effects, it will be worth analyzing typical forest species' occurrence versus ruderal species or dividing species into functional species groups. Maybe new, species-specific treatment effects could be found.
Discussion
In many results, there are differences between Bending Lake and E.B. Eddy sites. Why was there such a difference? Did some environmental conditions vary between sites that could affect results? Please add some information to the results or discussion section.
Author Response
General Response to Reviewer Comments
We are extremely appreciative of the enthusiastic comments and helpful suggestions of the reviewers. All suggestions to improve the manuscript have been considered, and specific replied to the individual comments by reviewers are included below.
The revised manuscript represents a long-term study of forest plantations focused on productivity and biodiversity. I appreciate the long-term observation that brings valuable results in assessing stand productivity and individual tree characteristics under different treatments. However, the weak point is the assessment of diversity indices. Since diversity observations were conducted 25 growing seasons after the treatments, I doubt the interpretation of results. The authors stated that treatments have a non-significant impact on diversity indices. In contrast, the impact on stand productivity was positive, which could be interpreted as all of the treatments are suitable management options even in the case of biodiversity protection. Since diversity sampling was not continuous, we do not know from the study the effect of treatments on diversity in few years after treatment. It is presumed that the effect of the treatment was very negative and could last for years. It is more likely that a non-significant impact on diversity was obtained due to vegetation's regeneration ability even after intensive treatment. Twenty-five years is enough time for vegetation to regenerate, so the results did not show that treatments have no impact on diversity. It only shows that after 25 years, vegetation can mitigate treatments' effect, but we have no information about processes between these years. I agree with the authors that trade-offs are needed in decision-making when forest management activities are planned. Ecosystem services have to be carefully assessed to quantify if more significant volumes after 25 years are worth 5, 10 maybe 15 years of disrupted forest ecosystem functions.
The results of the study are valuable and interesting from the forest production point of view. However, diversity measures have to be improved and carefully discussed.
Reply: We thank the reviewer for their thorough review of the manuscript. Concerns regarding specific comments have been responded to below.
Specific comments:
Introduction
Some basic descriptions of the vegetation communities' character that occur naturally in the area are missing, same as characteristic of jack pine plantations – what are the most widespread shrub species that use to be removed? How diverse these communities use to be?
Reply: The two studies were established under the Vegetation Management Alternatives Program (VMAP) post-harvest. Information about the post-harvest conditions are provided by Pitt et al. (2000) and Mallik et al. (2002).
Methods
A map of the study area with the location of plots could be added.
Reply: These plots have already been illustrated and described in details in previous publications (i.e., Pitt et al., 2000; Mallik et al., 2002). Moreover, we provide plot coordinates, a detailed description of the experimental design, and have included much information in Table 1 regarding the study areas. Based on this and the fact that the reviewer makes it a suggestion rather than a necessity, we believe that there is enough information on the study site and design that a map is not needed.
As stated in Methods, the study sites have different areas. Did the authors try to add site area as a random variable in models to test the influence?
Reply: We thank the reviewer for their concern regarding our statistical analyses. Site was included as a fixed variable in all statistical models, allowing us to explicitly test for differences in our variables between sties. Block was added as a random variable in the models to account for spatial variability within each site (please refer to Equation 1).
Diversity was assessed as the number and abundance of species and standard diversity indices. To better insight into treatment effects, it will be worth analyzing typical forest species' occurrence versus ruderal species or dividing species into functional species groups. Maybe new, species-specific treatment effects could be found.
Reply: Thank you for the suggestion. While it is indeed possible that the treatments altered the relative proportion of each species in the understorey, we did not observe any extirpations at the sites. Only one exotic species was able to colonize into the treated plots. The focus of this paper was to assess whether treatments led to uniform monocultures and we feel that the use of diversity indices tests this hypothesis sufficiently. However, we agree with the reviewer that a study focussed on the understorey community composition is warranted. To this end, we have a call to future studies to investigate shifts in understorey community traits in the Discussion (L 432-433).
Discussion
In many results, there are differences between Bending Lake and E.B. Eddy sites. Why was there such a difference? Did some environmental conditions vary between sites that could affect results? Please add some information to the results or discussion section.
Reply: We thank the reviewer for their comment. We believe we do a fair job in identifying site specific differences and treatment differences. We specifically address this in sections 4.1. and 4.2., in which we explain why there were significant differences between the Bending Lake and E.B. Eddy sites. For example, significantly lower conifer densities at E.B. Eddy (results; lines 271-272) were thought to be due to severe competition from overtopping aspen (discussion; lines 517-518), which was likely due to the fact that E.B Eddy was bladed, creating 3-4m wide berms covering 5-10% of the sample plot area, which acted as nutrient rich refugia for hardwood species (discussion; lines 357-358).